# Research on the Rotational Correction of Distributed Autonomous Orbit Determination in the Satellite Navigation Constellation

**Wei Zhou [1], Hongliang Cai [1], Ziqiang Li [2,\*], Chengpan Tang [3], Xiaogong Hu [3] and Wanke Liu [2]**

[1] Beijing Institute of Tracking and Telecommunications Technology (BITTT), 26 Beiqing Road, Beijing 100094, China; zhouwei_0611@163.com (W.Z.); caibanyu@126.com (H.C.)

[2] School of Geodesy and Geomatics, Wuhan University, Wuhan 430079, China; wkliu@sgg.whu.edu.cn

[3] Shanghai Astronomical Observatory, Chinese Academy of Sciences, Shanghai 200030, China; tcp@shao.ac.cn (C.T.); hxg@shao.ac.cn (X.H.)

\* Correspondence: ziqiangli@whu.edu.cn; Tel.: +86-15827546913

**Abstract:** The autonomous orbit determination of the navigation constellation uses only bidirectional ranging data of the inter-satellite link for data processing. The lack of space-time benchmark information related to the Earth inevitably causes overall rotational uncertainty in the constellation, leading to a decrease in orbit accuracy and affecting user positioning accuracy. This study (1) introduces a method for rotation correction in distributed autonomous orbit determination based on inter-satellite bidirectional ranging; (2) conducts constellation autonomous orbit determination and time synchronization processing experiments based on inter-satellite ranging data for the 24 medium Earth orbit (MEO) satellites in the Beidou-3 global satellite navigation system (BDS-3); and (3) makes comparative analyses on the accuracy of autonomous orbit determination based on three rotation correction cases, including a no-rotation-correction case, independent satellite constraints case, and global satellite constraints case. The experimental results are described as follows. For the no-rotation-correction case, the prediction error of the orbital inclination angle (iot, *i*) for the entire constellation on the 30th day was $2.11 \times 10^{-7}$/rad, the prediction error of the right ascension of the ascending point (Omega, $\Omega$) was $2.25 \times 10^{-7}$/rad, and the average root mean square (RMS) of the user range error (URE) for the entire constellation orbit was 1.41 m. In the autonomous orbit determination experiment with independent constraints on satellites, the prediction error of *i* for the entire constellation on the 30th day was $5.43 \times 10^{-7}$/rad, the prediction error of $\Omega$ was $2.03 \times 10^{-7}$/rad, and the average RMS of the orbital URE for the entire constellation was 1.09 m. In the autonomous orbit determination experiment with global satellite constraints, the prediction error of *i* for the entire constellation on the 30th day was $5.31 \times 10^{-7}$/rad, the prediction error of $\Omega$ was $1.95 \times 10^{-7}$/rad, and the RMS of the orbital URE for the entire constellation was 0.94 m. According to the analysis of the above experimental results, compared with the autonomous orbit determination under the no-rotation-correction case, the adoption of an algorithm for independent satellite constraints to correct the overall constellation rotation weakens the constellation rotation influence; however, it may destroy the overall constellation configuration, which affects the stability of autonomous orbit determination. Finally, the algorithm based on global satellite constraints both impairs the influence of constellation rotation and maintains the overall constellation configuration.

**Keywords:** Beidou satellite navigation system; distributed autonomous orbit determination; inter-satellite link; overall constellation rotation; algorithm for the correction of independent satellite constraints; algorithm for the correction of global satellite constraints

## 1. Introduction

In the 1980s, M.P. Ananda et al. proposed the concept of autonomous orbit determination of the navigation constellation [1]. This technology ensures the long-term,

autonomous, and stable operation and service capabilities of a navigation system, based on the distance measurement and communication functions of the inter-satellite link through the autonomous operation of the constellation and on-orbit ephemeris updates during long-term absence of ground system support. Subsequently, the initial theory, design, and data research work on autonomous orbit determination was conducted in June 1990 [2]. In July 2020, the Beidou-3 global satellite navigation system (BDS-3) began to operate and provide global service. The Ka-band inter-satellite link payload carried by the BDS-3 satellites can realize inter-satellite pseudorange measurement and inter-satellite communication, which enables research on the autonomous orbit determination of the constellation [3].

　　Autonomous navigation is first introduced for GPS Block IIR satellites and its design index requirement is that the user range error (URE) is less than 6 m within 180 days [4]. The autonomous orbit determination of navigation constellation uses only bidirectional ranging data of the inter-satellite link and lacks space-time benchmark information related to the Earth. Therefore, this technology cannot eliminate or suppress accumulation of the overall rotation errors of the constellation, making it difficult to operate autonomously for a long time [5–10]. Therefore, the design index requirement of GPS Block IIF satellites is changed to a URE of less than 2 m in a 60-day autonomous navigation [11]. Ananda et al. proved the unobservability of the overall rotation of the satellite constellation with inter-satellite measurements only and proposed an algorithm to constrain the right ascension of the ascending point by analyzing the long-predicted reference ephemeris provided by the Operational Control Segment (OCS) [12]. Abusali et al. pointed out that the orbit determination errors in the tangential and normal directions caused by the overall constellation rotation up to kilometer level, and the influence of errors in the predicted Earth Orientation Parameter (EOP), reach to meter level [13]. Gill et al. applied simulation data to study autonomous orbit determination [14]. With the development of BDS, several scholars researched the autonomous orbit determination technology of the navigation constellation by simulation, showing that the uncertainty of the right ascension of the ascending point $\Omega$ at the orbital plane will lead to overall rotation errors in orbit determination results [15–22]. In addition, other scholars also successively carried out related work, including algorithm research and simulation analysis [23–29]. The aforementioned studies aided in carrying out the research in this paper. This paper focuses on the problem of the overall constellation rotation correction in autonomous orbit determination, three correction cases are designed, and experiment results of different cases are analyzed and compared using inter-satellite observations.

## 2. Autonomous Orbit Determination Model

### 2.1. Observation Model for Autonomous Orbit Determination

　　Based on the concurrent spatial time division duplexing technology of the phased array antenna, bidirectional one-way distance measurement between the BDS-3 satellites was completed within 3.0 s [30]. At different times within these 3.0 s, the observation equations of the two satellites $SAT_A$ and $SAT_B$ can be expressed as.

$$\rho_{AB} = |R_B(t_1) - R_A(t_1 - \Delta t_1)| - \delta t_A + \delta t_B + c \cdot \tau_A^{send} + c \cdot \tau_B^{rec} + \Delta\rho_{AB} + \varepsilon_{AB} \quad (1)$$

$$\rho_{BA} = |R_A(t_2) - R_B(t_2 - \Delta t_2)| + \delta t_A - \delta t_B + c \cdot \tau_B^{send} + c \cdot \tau_A^{rec} + \Delta\rho_{BA} + \varepsilon_{BA} \quad (2)$$

where $R_A$ and $R_B$ are the positions of the two satellites $SAT_A$ and $SAT_B$, respectively; $c$ is the speed of light; $\delta t_A$ and $\delta t_B$ are the clock offsets of the two satellites $SAT_A$ and $SAT_B$, respectively; $\Delta t_1$ and $\Delta t_2$ are the propagation times of $\rho_{AB}$ and $\rho_{BA}$, respectively; $\tau^{send}$ and $\tau^{rec}$ are the emission delay and reception delay of the satellite, respectively; $\Delta\rho_{AB}$ and $\Delta\rho_{BA}$ are the error terms that can be modeled, including the relativistic effect and so on; and $\varepsilon_{AB}$ and $\varepsilon_{BA}$ are the noises of the respective observed values.

　　In addition, it is necessary to further reduce the inter-satellite bidirectional observations at different times to the same time to participate in the satellite orbit determination. First, utilizing the predicted orbit and the predicted satellite clock offset parameters, the observed values measured at $t_1$ and $t_2$ are reduced to the nearest full 3 s at time $t_0$. Considering that

the bidirectional observed values are completed in a very short time, the prediction errors for the orbit and the clock offset can be ignored.

$$\rho_{AB}(t_0) = \rho_{AB} + |R_B(t_0) - R_A(t_0)| - |R_B(t_1) - R_A(t_1 - \Delta t_1)| + \\ c \cdot (\delta t_B(t_0) - \delta t_A(t_0)) - c \cdot (\delta t_A(t_1) - \delta t_B(t_1 - \Delta t_1)) \tag{3}$$

$$\rho_{BA}(t_0) = \rho_{BA} + |R_A(t_0) - R_B(t_0)| - |R_A(t_2) - R_B(t_2 - \Delta t_2)| + \\ c \cdot (\delta t_A(t_0) - \delta t_B(t_0)) - c \cdot (\delta t_A(t_2) - \delta t_B(t_2 - \Delta t_2)) \tag{4}$$

Then, adding the bidirectional pseudorange at time $t_0$, the observation equation that only contains the satellite orbit parameters can be obtained by sorting

$$\frac{\rho_{AB}(t_0) + \rho_{BA}(t_0)}{2} = |R_B(t_0) - R_A(t_0)| + \frac{c \cdot \tau_A^+}{2} + \frac{c \cdot \tau_B^+}{2} + \varepsilon^+ \tag{5}$$

The above equation is the orbit determination observation equation in autonomous orbit determination, where $\tau^+$ is the sum of the emission delay and reception delay of the satellite, which is referred to as the time delay sum parameter in autonomous orbit determination.

In this contribution, we adopt an extended Kalman filter (EKF) to estimate the satellite's orbit parameters to ensure the real-time performance, in each of which we simultaneously estimate all satellite orbit parameters in a distribute processing mode.

### 2.2. Rotation Correction Model

In autonomous orbit determination based on only inter-satellite bidirectional ranging, the state parameters of all satellites in the constellation must be estimated, which means a rank deficiency for the solution of the parameters and the lack of the necessary starting benchmark. Thus, it is impossible to determine the absolute positions of all the satellites. In autonomous orbit determination, the observed and real inter-satellite distances of the constellation are the same, meaning the constellation estimation errors cannot be observed theoretically. The unobserved constellation estimation errors of autonomous orbit determination by inter-satellite ranging measurements mainly refer to the unobservability of constellation rotation; on one hand, inter-satellite ranging cannot correct some of the orbital elements and, on the other hand, the unobservability is brought about by using the prediction of EOP [27].

In the current distributed autonomous orbit determination, the two orbital plane orientation parameters of the right ascension for the ascending point $\Omega$ and the orbital inclination angle *i* are adopted as the constraint conditions to limit the overall constellation rotation satellite specifically, which is the algorithm for the correction of independent satellite constraints. Since the effect (angle) of the constellation rotation error on each satellite is consistent, the independent constraint algorithm eliminates this consistency; therefore, the result is suboptimal. After estimating and obtaining the constellation rotation error of the orbit solved by filtering relative to the reference orbit, the rotation correction algorithm for the global satellite constraints can then directly correct the orbit parameters obtained in autonomous orbit determination to realize the suppression of the overall rotation error of the constellation, which guarantees the integrity of the constellation.

2.2.1. Theoretical Analysis of the Influence of the Overall Constellation Rotation

At a certain time, the position and velocity of a certain satellite in the constellation are

$$X = \begin{pmatrix} x & y & z & \dot{x} & \dot{y} & \dot{z} \end{pmatrix}^T \tag{6}$$

If there is an overall rotation error with a rotation quantity of $\theta(\theta_\alpha, \theta_\beta, \theta_\gamma)$ in the constellation for autonomous orbit determination, then according to the criteria for coordinate transformation, the estimated value $X'$ of the satellite position state is

$$X' = \begin{pmatrix} x' \\ y' \\ z' \\ \dot{x}' \\ \dot{y}' \\ \dot{z}' \end{pmatrix} = \begin{bmatrix} R(-\theta) & 0 \\ 0 & R(-\theta) \end{bmatrix} \begin{pmatrix} x \\ y \\ z \\ \dot{x} \\ \dot{y} \\ \dot{z} \end{pmatrix} = R^* X \tag{7}$$

where $R$ represents the rotation matrix of the constellation; then,

$$R(-\theta) = R_\gamma(-\theta_\gamma) R_\beta(-\theta_\beta) R_\alpha(-\theta_\alpha)$$

$$R_\alpha(-\theta_\alpha) = \begin{bmatrix} 1 & 0 & 0 \\ 0 & cos\theta_\alpha & -sin\theta_\alpha \\ 0 & sin\theta_\alpha & cos\theta_\alpha \end{bmatrix}$$

$$R_\beta(-\theta_\beta) = \begin{bmatrix} cos\theta_\beta & 0 & sin\theta_\beta \\ 0 & 1 & 0 \\ -sin\theta_\beta & 0 & cos\theta_\beta \end{bmatrix}$$

$$R_\gamma(-\theta_\gamma) = \begin{bmatrix} cos\theta_\gamma & -sin\theta_\gamma & 0 \\ sin\theta_\gamma & cos\theta_\gamma & 0 \\ 0 & 0 & 1 \end{bmatrix}$$

where $R_\alpha$, $R_\beta$, and $R_\gamma$ represent the rotation matrices around the X-axis, Y-axis, and Z-axis, respectively (same below). When $\theta$ is a small quantity, we have

$$R(-\theta) \approx \begin{bmatrix} 1 & -\theta_\gamma & \theta_\beta \\ \theta_\gamma & 1 & -\theta_\alpha \\ -\theta_\beta & \theta_\alpha & 1 \end{bmatrix} \tag{8}$$

Therefore,

$$R^* \approx \begin{bmatrix} 1 & -\theta_\gamma & \theta_\beta & 0 & 0 & 0 \\ \theta_\gamma & 1 & -\theta_\alpha & 0 & 0 & 0 \\ -\theta_\beta & \theta_\alpha & 1 & 0 & 0 & 0 \\ 0 & 0 & 0 & 1 & -\theta_\gamma & \theta_\beta \\ 0 & 0 & 0 & \theta_\gamma & 1 & -\theta_\alpha \\ 0 & 0 & 0 & -\theta_\beta & \theta_\alpha & 1 \end{bmatrix} \tag{9}$$

Noting that

$$X' = R^* X \approx \begin{pmatrix} x + z\theta_\beta - y\theta_\gamma \\ y - z\theta_\alpha + x\theta_\gamma \\ z + y\theta_\alpha - x\theta_\beta \\ \dot{x} + \dot{z}\theta_\beta - \dot{y}\theta_\gamma \\ \dot{y} - \dot{z}\theta_\alpha + \dot{x}\theta_\gamma \\ \dot{z} + \dot{y}\theta_\alpha - \dot{x}\theta_\beta \end{pmatrix} = \begin{pmatrix} x \\ y \\ z \\ \dot{x} \\ \dot{y} \\ \dot{z} \end{pmatrix} + \begin{bmatrix} 0 & z & -y \\ -z & 0 & x \\ y & -x & 0 \\ 0 & \dot{z} & -\dot{y} \\ -\dot{z} & 0 & \dot{x} \\ \dot{y} & -\dot{x} & 0 \end{bmatrix} \begin{pmatrix} \theta_\alpha \\ \theta_\beta \\ \theta_\gamma \end{pmatrix} \tag{10}$$

$$H = \begin{bmatrix} 0 & z & -y \\ -z & 0 & x \\ y & -x & 0 \\ 0 & \dot{z} & -\dot{y} \\ -\dot{z} & 0 & \dot{x} \\ \dot{y} & -\dot{x} & 0 \end{bmatrix} \tag{11}$$

$$X' \approx X + H\theta \tag{12}$$

That is, the satellite position and velocity state errors caused by the constellation rotation error are

$$\delta X = X' - X \approx H\theta = \begin{pmatrix} z\theta_\beta - y\theta_\gamma \\ x\theta_\gamma - z\theta_\alpha \\ y\theta_\alpha - x\theta_\beta \\ \dot{z}\theta_\beta - \dot{y}\theta_\gamma \\ \dot{x}\theta_\gamma - \dot{z}\theta_\alpha \\ \dot{y}\theta_\alpha - \dot{x}\theta_\beta \end{pmatrix} \tag{13}$$

### 2.2.2. Algorithm for Independent Satellite Constraints

The constraint conditions for the algorithm for independent satellite constraints are shown as follows:

$$\begin{cases} i = \tilde{i} \\ \Omega = \widetilde{\Omega} \end{cases} \tag{14}$$

where $\tilde{i}$ and $\widetilde{\Omega}$ are the orbital inclination angle and the right ascension of the ascending point of the reference orbit, respectively, and $i$ and $\Omega$ are the corresponding parameters to be estimated. Considering $i$ and $\Omega$ as functions of state $X$ to be estimated (Cartesian coordinates and velocity in the inertial system) and linearly expanding at the approximate value $X_0$, we have

$$\begin{cases} i_0 + \left.\frac{\partial i}{\partial X}\right|_{X_0} (X - X_0) = \tilde{i} \\ \Omega_0 + \left.\frac{\partial \Omega}{\partial X}\right|_{X_0} (X - X_0) = \widetilde{\Omega} \end{cases} \tag{15}$$

where $i_0$ and $\Omega_0$ are obtained by calculation based on $X_0$, and

$$\begin{bmatrix} \frac{\partial i}{\partial X} \\ \frac{\partial \Omega}{\partial X} \end{bmatrix} = \begin{bmatrix} \frac{\partial i}{\partial \vec{r}} & \frac{\partial i}{\partial \dot{\vec{r}}} \\ \frac{\partial \Omega}{\partial \vec{r}} & \frac{\partial \Omega}{\partial \dot{\vec{r}}} \end{bmatrix} = \begin{bmatrix} \frac{\partial i}{\partial x} & \frac{\partial i}{\partial y} & \frac{\partial i}{\partial z} & \frac{\partial i}{\partial \dot{x}} & \frac{\partial i}{\partial \dot{y}} & \frac{\partial i}{\partial \dot{z}} \\ \frac{\partial \Omega}{\partial x} & \frac{\partial \Omega}{\partial y} & \frac{\partial \Omega}{\partial z} & \frac{\partial \Omega}{\partial \dot{x}} & \frac{\partial \Omega}{\partial \dot{y}} & \frac{\partial \Omega}{\partial \dot{z}} \end{bmatrix} \tag{16}$$

When the one-step prediction value from the filtering results of the previous epoch to the start time of the current epoch is selected as the linearized expansion point $X_0$, Equation (16) can be written as a general expression form of the constraint equation

$$C\delta X = M \tag{17}$$

where

$$C = \begin{bmatrix} \frac{\partial i}{\partial X} \\ \frac{\partial \Omega}{\partial X} \end{bmatrix}_{X_0}, \; M = \begin{pmatrix} \tilde{i} - i_0 \\ \widetilde{\Omega} - \Omega_0 \end{pmatrix}$$

Therefore, the complete orbit filtering model of the algorithm for independent constraints on satellites in distributed autonomous orbit determination is

$$\begin{cases} \delta X_k = \Phi_{k,k-1}\delta X_{k-1} + W_{k-1} \\ Z_k = H_k \delta X_k + Y_k + V_k \\ C_k \delta X_k = M_k \end{cases} \tag{18}$$

### 2.2.3. Algorithm for Global Satellite Constraints

The estimated value of the inclination angle $i$ and the right ascension of the ascending point $\Omega$ in the orbit of the orbit elements obtained in the autonomous orbit determination of Satellite $m$ is $\sigma'_m(i'_m, \Omega'_m)$, and the estimated value of the inclination angle $i$ and the right ascension of the ascending point $\Omega$ among the real orbital elements of the satellite is $\sigma_m(i_m, \Omega_m)$. Due to the unobservable satellites of constellation rotation, when an overall

rotation of a rotation quantity $\theta\left(\theta_\alpha, \theta_\beta, \theta_\gamma\right)$ occurs, the following relationship will exist between $\sigma'_m$ and $\sigma_m$:

$$\sigma = \widetilde{\sigma} + H\theta \tag{19}$$

In the above equation, the partial derivative matrix of $H$ Kepler orbital roots to the constellation rotation error is given without additional derivation:

$$H = \frac{\partial \sigma}{\partial \theta} = \begin{pmatrix} \frac{\partial i}{\partial \theta_\alpha} & \frac{\partial i}{\partial \theta_\beta} & \frac{\partial i}{\partial \theta_\gamma} \\ \frac{\partial \Omega}{\partial \theta_\alpha} & \frac{\partial \Omega}{\partial \theta_\beta} & \frac{\partial \Omega}{\partial \theta_\gamma} \end{pmatrix} = \begin{pmatrix} \cos \Omega_m & \sin \Omega_m & 0 \\ -\sin \Omega_m \cot i_m & \cos \Omega_m \cot i_m & 1 \end{pmatrix} \tag{20}$$

To estimate the rotation quantity $\theta\left(\theta_\alpha, \theta_\beta, \theta_\gamma\right)$, the errors in the inclination angle $i$ and in the right ascension of the ascending point $\Omega$ of at least two satellite orbital elements are needed. In fact, in estimating a more optimal constellation rotation quantity $\theta$, it is smaller the more satellites there are, and these satellites should be distributed as evenly as possible on each orbital plane. If the errors in the inclination angle $i$ and in the right ascension of the ascending point $\Omega$ of the left and right satellites in the constellation are known, and Equation (19)of each satellite is listed, we have

$$U = X - H\theta \tag{21}$$

In the equation, $U = (U_1, U_2, \ldots, U_m, \ldots, U_n)^T$, $H = (H_1, H_2, \ldots, H_m, \ldots, H_n)^T$, and $U_m = \sigma'_m - \sigma_m - H_m\theta$. The optimal estimated value of $\theta$ can be determined under the least squares criterion.

$$\hat{\theta} = \left(H^T P H\right)^{-1} H^T P X \tag{22}$$

After estimating and obtaining the constellation rotation error of the orbit solved by filtering relative to the reference orbit, the results obtained in autonomous orbit determination can be directly corrected.

## 3. Analysis of the Inter-Satellite Measurement Situation

### 3.1. Analysis of the Link Establishment Situation for Inter-Satellite Links

According to the constellation configuration, there are three kinds of visible relationships among BDS-3 MEO satellites, namely, continuously visible, non-continuously visible, and invisible, which are related to the continuity of inter-satellite measurement, the number of established links, the orbital positions of satellites, and time slot route planning, etc. Z.L., J.X. et al. analyzed the inter-satellite observation conditions [30]. Figure 1 shows the 30-day time series of the number of satellites with established links for the PRN36 satellite. The fluctuation is considerable, with a minimum of 0 and a maximum of up to 20. Table 1 shows the statistics for the number of established links. In general, the minimum number of established links is 0, the maximum is 19.96, and the average number of established links is 14.45.

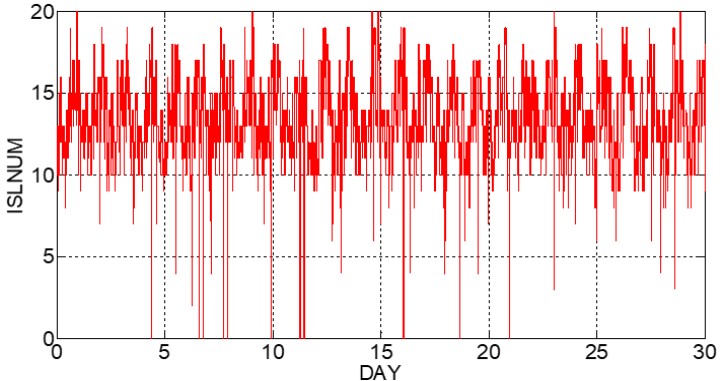

**Figure 1.** The 30-day link-tracking number of PRN36 for BDS-3.

**Table 1.** The statistical situation in the number of established links for each BDS-3 satellite.

| PRN | Maximum Number of Established Links | Minimum Number of Established Links | Mean |
|---|---|---|---|
| 25 | 20 | 0 | 14.55 |
| 26 | 20 | 0 | 14.55 |
| 27 | 20 | 0 | 14.41 |
| 28 | 20 | 0 | 14.25 |
| 29 | 20 | 0 | 14.20 |
| 30 | 20 | 0 | 13.68 |
| 19 | 20 | 0 | 14.56 |
| 20 | 20 | 0 | 14.48 |
| 21 | 20 | 0 | 14.41 |
| 22 | 20 | 0 | 14.73 |
| 23 | 20 | 0 | 14.97 |
| 24 | 20 | 0 | 14.95 |
| 32 | 20 | 0 | 15.06 |
| 33 | 19 | 0 | 13.98 |
| 34 | 20 | 0 | 13.92 |
| 35 | 20 | 0 | 14.21 |
| 36 | 20 | 0 | 14.45 |
| 37 | 20 | 0 | 14.58 |
| 38 | 20 | 0 | 14.66 |
| 39 | 20 | 0 | 14.80 |
| 40 | 20 | 0 | 14.01 |
| 41 | 20 | 0 | 14.18 |
| 42 | 20 | 0 | 14.91 |
| 43 | 20 | 0 | 14.09 |
| Mean | 19.96 | 0 | 14.45 |

*3.2. Constellation Configuration Analysis for Autonomous Orbit Determination*

When the number of established links between satellites to be assessed is greater than or equal to three, the corresponding position dilution of precision (PDOP) value can be calculated to reflect the quality of the geometric configuration of inter-satellite link establishment. Figure 2 presents the changing situation for the PDOP values of the PRN36 satellite during a consecutive 30-day period, and Table 2 shows the PDOP statistics of all 24 MEO satellites in BDS-3. The average minimum PDOP of all satellites is approximately 0.74, the average maximum PDOP is approximately 4.41, and the mean PDOP value is approximately 0.99. It can be concluded that the overall structure of the geometric figure for inter-satellite link establishment is improved [26].

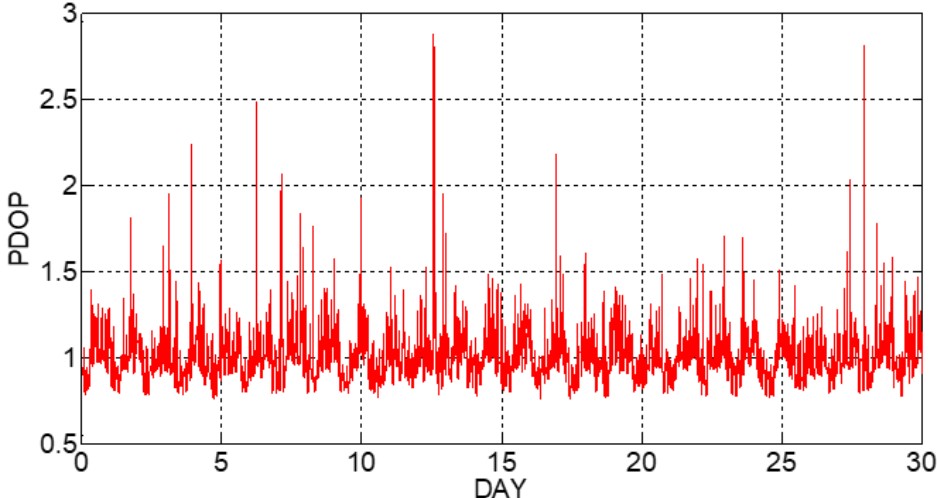

**Figure 2.** The changing situation of PDOP values for PRN36 satellites as the positioning objects of BDS-3.

**Table 2.** The statistical situation of PDOP values for the BDS-3 satellites.

| PRN | Minimum | Maximum | Mean |
|---|---|---|---|
| 25 | 0.75 | 3.33 | 0.94 |
| 26 | 0.73 | 4.59 | 0.95 |
| 27 | 0.73 | 3.93 | 0.96 |
| 28 | 0.74 | 3.54 | 0.97 |
| 29 | 0.73 | 4.50 | 0.96 |
| 30 | 0.73 | 4.04 | 0.98 |
| 19 | 0.75 | 2.89 | 0.95 |
| 20 | 0.73 | 5.57 | 0.95 |
| 21 | 0.75 | 5.31 | 0.96 |
| 22 | 0.74 | 4.95 | 0.96 |
| 23 | 0.74 | 4.23 | 0.94 |
| 24 | 0.74 | 4.63 | 0.94 |
| 32 | 0.73 | 3.07 | 0.94 |
| 33 | 0.75 | 3.39 | 0.98 |
| 34 | 0.74 | 5.64 | 0.98 |
| 35 | 0.73 | 4.71 | 0.96 |
| 36 | 0.75 | 2.91 | 0.96 |
| 37 | 0.75 | 5.96 | 0.95 |
| 38 | 0.73 | 4.42 | 0.94 |
| 39 | 0.74 | 5.70 | 0.94 |
| 40 | 0.74 | 4.04 | 1.06 |
| 41 | 0.75 | 4.21 | 1.77 |
| 42 | 0.75 | 3.93 | 0.94 |
| 43 | 0.75 | 4.62 | 0.97 |
| Mean | 0.74 | 4.41 | 0.99 |

## 4. Orbit Determination Results

### 4.1. Processing Cases and Strategies for Autonomous Orbit Determination

Distributed autonomous orbit determination processing was adopted in this paper to assess the accuracy of the autonomous orbit determination of BDS-3 under different rotation correction methods.

In the general processing of precise orbit determination, parameters to be estimated include initial satellite position and velocity, solar radiation pressure parameters, clock offset parameters (clock offset and clock speed), as well as other parameters induced by measurements. Targeted optimization was carried out on the autonomous orbit determination algorithm in this paper. Table 3 shows the specific models and relevant strategies adopted in orbit determination. Among them, the Empirical CODE orbit Model (ECOM) was adopted for modeling solar radiation pressure perturbation, and a longer arc segment of the satellite-Earth-satellite joint orbit determination result was used to estimate the solar radiation pressure parameters more precisely. In this way, solar radiation pressure parameters were no longer estimated, thereby ensuring fewer estimated parameters and a smaller computational load.

### 4.2. Data Processing Strategies for Autonomous Orbit Determination

In data processing for autonomous orbit determination in this paper, necessary correction of observation was first carried out. Then, time reduction processing was carried out according to the predicted orbit of the current epoch. Afterwards, gross errors in observations were detected and eliminated and high-quality observations were acquired. Finally, the autonomous orbit determination was conducted and the data processing flow is shown in Figure 3.

**Table 3.** Processing strategies of autonomous orbit determination (AOD).

| Parameter | Model |
|---|---|
| Observed values | Observation data of inter-satellite links |
| Observation interval | 1 min |
| Satellite transceiver delay | Not estimated, calibrated numerical values are adopted |
| Gravity field model | Earth Gravitational Model 2008 (EGM2008) model to the 8th order |
| Tidal correction | Only solid tides are considered |
| Solar radiation pressure model | ECOM model, parameters not estimated |
| EOP | International Earth Rotation and Reference Systems Service (IERS) prediction of EOP (Bulletin A) |
| Gravitational force of N body | Considers the gravitational forces of the sun and moon |
| Parameters of the initial orbit | Broadcast ephemeris orbit |
| Parameters of the initial clock offset | Broadcast ephemeris clock offset |
| Estimator | Extended Kalman filter (EKF) |
| Parameters to be estimated | Only the position and velocity parameters of each satellite are estimated |

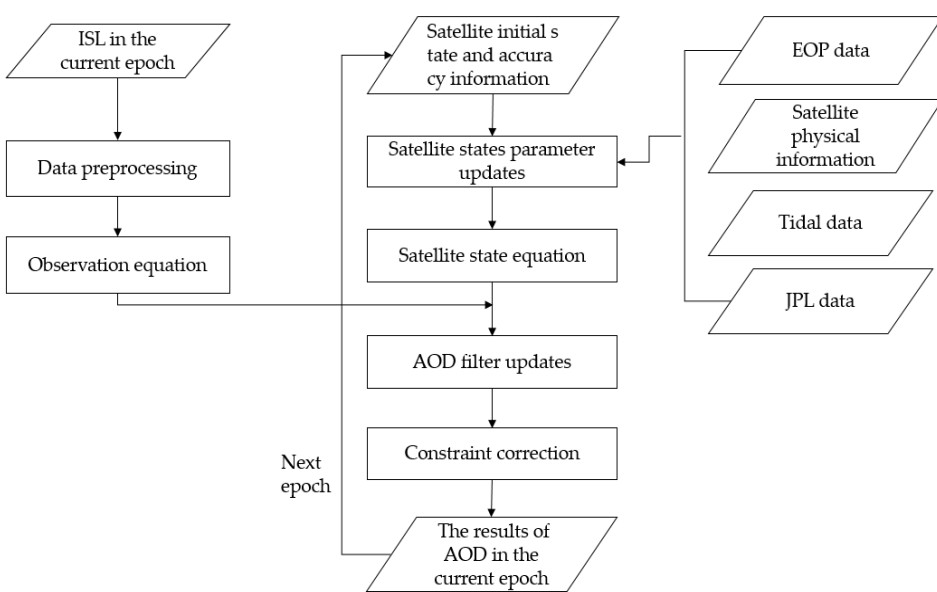

**Figure 3.** AOD data processing flowchart.

*4.3. Analysis of Experimental Results*

　　To verify the effect of the aforementioned rotation correction methods on the accuracy of autonomous orbit determination solutions, the following three kinds of a priori constraint information were adopted in data filtering processing.

　　Case 1: No rotation correction.

　　Case 2: Adoption of an algorithm for independent constraints to correct the rotation.

　　Case 3: Adoption of an algorithm for overall constraints onboard the satellite to correct the rotation.

　　A priori constraint information was generated for the three aforementioned cases, respectively, and autonomous orbit determination simulation processing was carried out with inter-satellite ranging data from 16 October 2020 to 14 November 2020 of 24 MEO satellites of BDS-3.

Using the satellite-ground and inter-satellite joint orbit determination results as reference, the rotation errors acquired by Case 1, Case 2, and Case 3 were analyzed. The three Euler angles (α, β, γ) of the constellation rotation parameters were calculated, and the results are presented in Figure 4. Figure 5 shows the prediction errors of UT1-UTC in EOP. Compared with the no-rotation-correction case, both the algorithm for independent constraints and the algorithm for global constraints could constrain the constellation rotation on the X and Y axes better, while no significant improvement was shown on the Z-axis. Moreover, the errors were basically consistent with the prediction errors of UT1-UTC in EOP given in Figure 6, which occurred because the errors on the Z-axis in the overall rotation of the constellation were mainly caused by the prediction errors of UT1-UTC, and the errors could not be eliminated without external reference input.

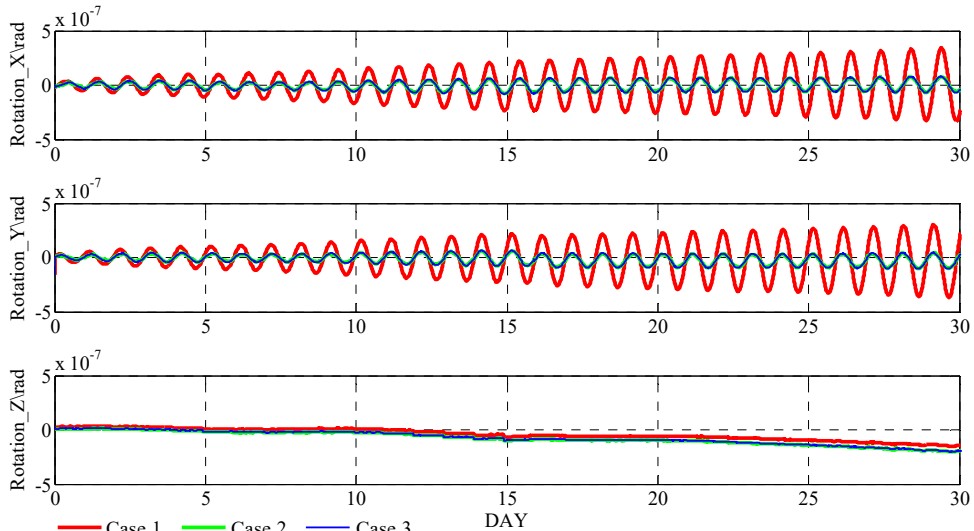

**Figure 4.** Euler angles of the constellation rotation in each direction under different autonomous orbit determination cases.

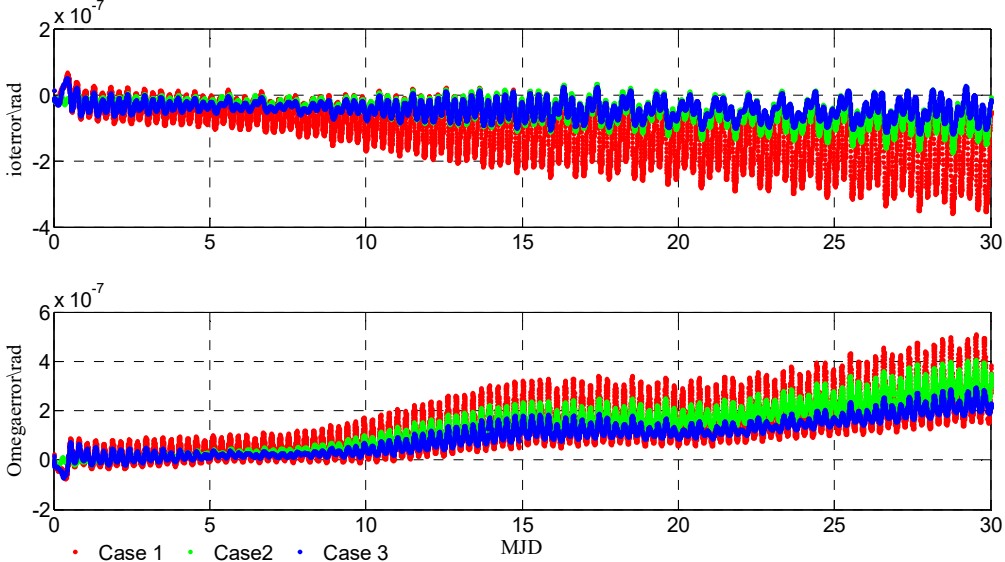

**Figure 5.** The *i* angle and *Ω* angle errors of PRN25 under different cases for autonomous orbit determination.

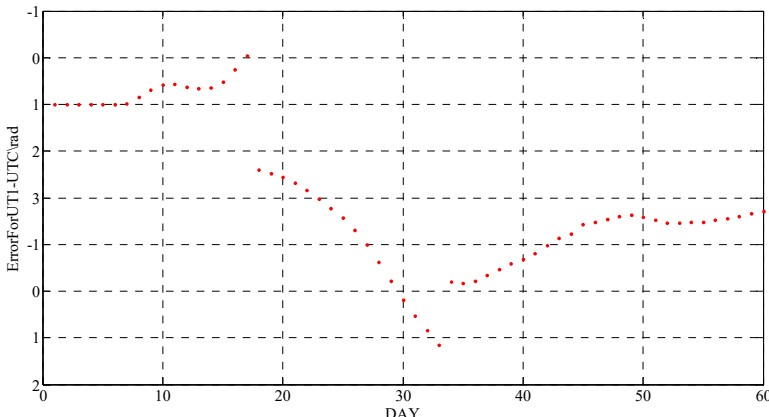

**Figure 6.** 30-day prediction errors of UT1-UTC in EOP.

Furthermore, the orbit inclination angle *i* (iot) and right ascension of the ascending point $\Omega$ (Omega) under the three cases were calculated. Using the post-processed precise orbits of satellite-ground and inter-satellite joint orbit determination as references, the angle errors relative to the precise ephemeris error were obtained. Due to the space limitations of the article, only the time series of the PRN25 and PRN37 satellites are given here, as shown in Figures 5 and 7. Table 4 shows the prediction errors in the *i* and $\Omega$ for each satellite on the 30th day under the different cases. It can be seen from Figures 7 and 8, when the rotation was not corrected, the *i* and $\Omega$ of PRN25 and PRN37 became divergent gradually, and the trends of the two satellites were basically consistent. After adopting the algorithm for independent satellite constraints, the *i* errors of the two satellites could be better constrained, and there was also a correction effect on $\Omega$. However, the margin of error for each satellite after correction was inconsistent, which affected the integrity of the constellation as a rigid body. The algorithm for global constraints on satellites corrected the *i* and $\Omega$ well, and the errors after correction were basically consistent, which did not affect the integrity of the constellation as a rigid body. Without any correction, the average RMS of the *i* prediction errors for the entire constellation on the 30th day was $2.11 \times 10^{-7}$/rad, and the average RMS of the $\Omega$ prediction errors was $2.25 \times 10^{-7}$/rad. After adopting the algorithm for independent satellite constraints, the *i* prediction error was $5.43 \times 10^{-8}$/rad, and the $\Omega$ prediction error was $2.03 \times 10^{-7}$/rad. When the algorithm for global satellite constraints was applied, the average *i* prediction error was $5.31 \times 10^{-8}$/rad, and the average $\Omega$ prediction error was $1.95 \times 10^{-7}$/rad. The two constraint algorithms could weaken the *i* and $\Omega$ prediction errors of the constellation to some extent, and the algorithm for global satellite constraints achieved the best performance.

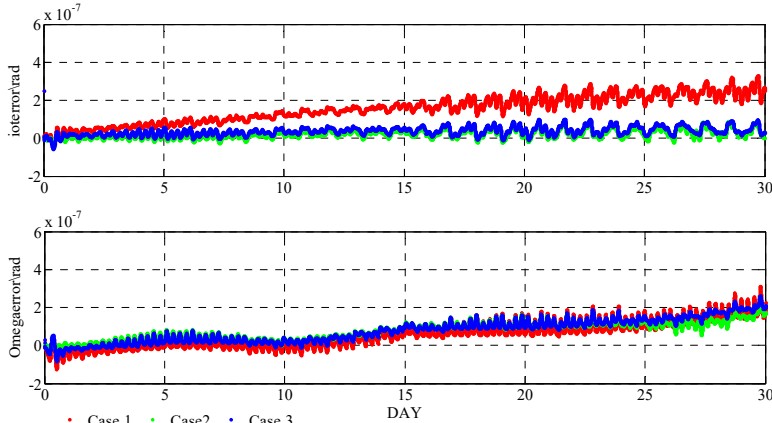

**Figure 7.** The *i* angle and $\Omega$ angle errors of PRN37 under different cases for autonomous orbit determination.

**Table 4.** The *i* and *Ω* prediction errors on the 30th day of autonomous orbit determination under different cases.

| PRN | *i*/rad | | | *Ω*/rad | | |
|---|---|---|---|---|---|---|
| | Case 1 | Case 2 | Case 3 | Case 1 | Case 2 | Case 3 |
| 25 | $1.99 \times 10^{-7}$ | $2.86 \times 10^{-8}$ | $4.65 \times 10^{-8}$ | $3.07 \times 10^{-7}$ | $1.94 \times 10^{-7}$ | $2.23 \times 10^{-7}$ |
| 26 | $2.19 \times 10^{-7}$ | $3.70 \times 10^{-8}$ | $5.33 \times 10^{-8}$ | $3.06 \times 10^{-7}$ | $2.17 \times 10^{-7}$ | $2.29 \times 10^{-7}$ |
| 27 | $2.65 \times 10^{-7}$ | $5.72 \times 10^{-8}$ | $6.06 \times 10^{-8}$ | $1.75 \times 10^{-7}$ | $1.59 \times 10^{-7}$ | $1.92 \times 10^{-7}$ |
| 28 | $2.65 \times 10^{-7}$ | $5.83 \times 10^{-8}$ | $6.14 \times 10^{-8}$ | $1.80 \times 10^{-7}$ | $2.03 \times 10^{-7}$ | $1.92 \times 10^{-7}$ |
| 29 | $2.53 \times 10^{-7}$ | $4.16 \times 10^{-8}$ | $5.43 \times 10^{-8}$ | $1.97 \times 10^{-7}$ | $1.62 \times 10^{-7}$ | $2.03 \times 10^{-7}$ |
| 30 | $2.57 \times 10^{-7}$ | $4.91 \times 10^{-8}$ | $5.77 \times 10^{-8}$ | $1.84 \times 10^{-7}$ | $1.26 \times 10^{-7}$ | $1.93 \times 10^{-7}$ |
| 19 | $1.59 \times 10^{-7}$ | $2.98 \times 10^{-8}$ | $4.83 \times 10^{-8}$ | $1.41 \times 10^{-7}$ | $1.63 \times 10^{-7}$ | $1.30 \times 10^{-7}$ |
| 20 | $1.58 \times 10^{-7}$ | $4.32 \times 10^{-8}$ | $4.46 \times 10^{-8}$ | $1.49 \times 10^{-7}$ | $1.78 \times 10^{-7}$ | $1.52 \times 10^{-7}$ |
| 21 | $1.59 \times 10^{-7}$ | $5.99 \times 10^{-8}$ | $4.49 \times 10^{-8}$ | $1.50 \times 10^{-7}$ | $1.28 \times 10^{-7}$ | $1.57 \times 10^{-7}$ |
| 22 | $1.52 \times 10^{-7}$ | $4.16 \times 10^{-8}$ | $4.62 \times 10^{-8}$ | $1.51 \times 10^{-7}$ | $1.35 \times 10^{-7}$ | $1.30 \times 10^{-7}$ |
| 23 | $2.04 \times 10^{-7}$ | $4.09 \times 10^{-8}$ | $5.07 \times 10^{-8}$ | $2.96 \times 10^{-7}$ | $2.11 \times 10^{-7}$ | $2.17 \times 10^{-7}$ |
| 24 | $2.12 \times 10^{-7}$ | $4.39 \times 10^{-8}$ | $5.18 \times 10^{-8}$ | $3.21 \times 10^{-7}$ | $2.30 \times 10^{-7}$ | $2.32 \times 10^{-7}$ |
| 32 | $1.74 \times 10^{-7}$ | $6.73 \times 10^{-8}$ | $5.98 \times 10^{-8}$ | $1.65 \times 10^{-7}$ | $1.58 \times 10^{-7}$ | $1.48 \times 10^{-7}$ |
| 33 | $1.55 \times 10^{-7}$ | $5.43 \times 10^{-8}$ | $4.46 \times 10^{-8}$ | $1.41 \times 10^{-7}$ | $1.58 \times 10^{-7}$ | $1.58 \times 10^{-7}$ |
| 34 | $2.59 \times 10^{-7}$ | $3.52 \times 10^{-8}$ | $5.92 \times 10^{-8}$ | $1.89 \times 10^{-7}$ | $2.55 \times 10^{-7}$ | $1.99 \times 10^{-7}$ |
| 35 | $2.61 \times 10^{-7}$ | $5.24 \times 10^{-8}$ | $5.82 \times 10^{-8}$ | $1.97 \times 10^{-7}$ | $2.23 \times 10^{-7}$ | $2.05 \times 10^{-7}$ |
| 36 | $2.02 \times 10^{-7}$ | $5.66 \times 10^{-8}$ | $5.13 \times 10^{-8}$ | $3.16 \times 10^{-7}$ | $2.00 \times 10^{-7}$ | $2.29 \times 10^{-7}$ |
| 37 | $2.22 \times 10^{-7}$ | $5.07 \times 10^{-8}$ | $5.70 \times 10^{-8}$ | $3.02 \times 10^{-7}$ | $1.92 \times 10^{-7}$ | $2.22 \times 10^{-7}$ |
| 41 | $1.50 \times 10^{-7}$ | $5.84 \times 10^{-8}$ | $4.47 \times 10^{-8}$ | $1.53 \times 10^{-7}$ | $1.43 \times 10^{-7}$ | $1.50 \times 10^{-7}$ |
| 42 | $1.58 \times 10^{-7}$ | $5.48 \times 10^{-8}$ | $4.31 \times 10^{-8}$ | $1.50 \times 10^{-7}$ | $1.58 \times 10^{-7}$ | $1.56 \times 10^{-7}$ |
| 43 | $2.58 \times 10^{-7}$ | $4.47 \times 10^{-8}$ | $5.91 \times 10^{-8}$ | $1.94 \times 10^{-7}$ | $2.48 \times 10^{-7}$ | $2.01 \times 10^{-7}$ |
| 44 | $2.64 \times 10^{-7}$ | $5.95 \times 10^{-8}$ | $6.02 \times 10^{-8}$ | $1.87 \times 10^{-7}$ | $2.23 \times 10^{-7}$ | $1.99 \times 10^{-7}$ |
| 45 | $2.08 \times 10^{-7}$ | $9.51 \times 10^{-8}$ | $5.37 \times 10^{-8}$ | $2.98 \times 10^{-7}$ | $3.24 \times 10^{-7}$ | $2.25 \times 10^{-7}$ |
| 46 | $2.17 \times 10^{-7}$ | $8.82 \times 10^{-8}$ | $5.66 \times 10^{-8}$ | $3.19 \times 10^{-7}$ | $3.12 \times 10^{-7}$ | $2.31 \times 10^{-7}$ |
| RMS | $2.11 \times 10^{-7}$ | $5.43 \times 10^{-8}$ | $5.31 \times 10^{-8}$ | $2.25 \times 10^{-7}$ | $2.03 \times 10^{-7}$ | $1.95 \times 10^{-7}$ |
| STD | $1.58 \times 10^{-7}$ | $1.55 \times 10^{-8}$ | $6.04 \times 10^{-9}$ | $6.78 \times 10^{-7}$ | $5.15 \times 10^{-8}$ | $2.91 \times 10^{-8}$ |

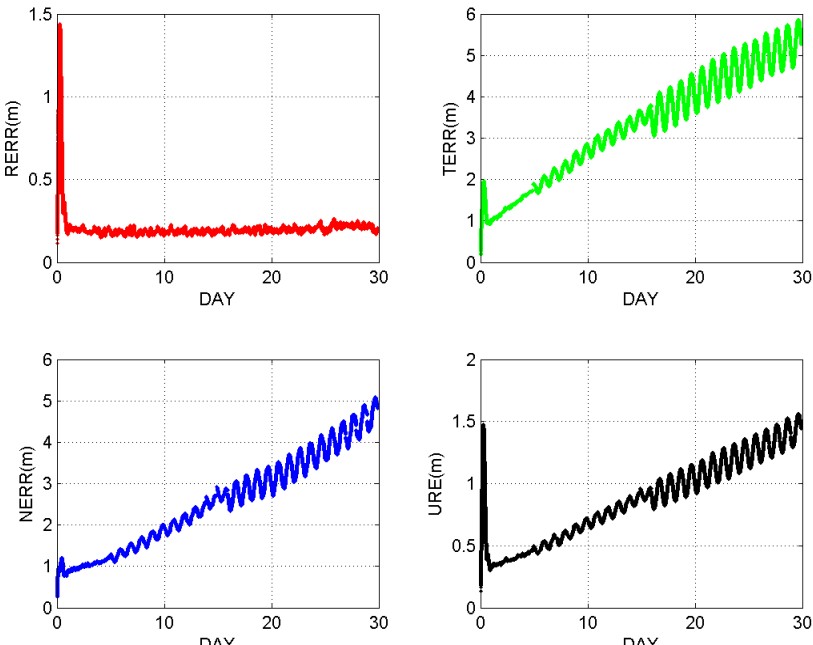

**Figure 8.** The autonomous navigation constellation average orbit error with no-rotation-correction case.

The differences between the autonomous orbit determination results and the reference orbits were analyzed, and the radial errors (RERR), tangential errors (TERR), normal-

direction errors (NERR), and comprehensive errors (user range errors, URE) of the constellation are given in Figures 8–10. The 30-day accuracy statistics for autonomous orbit determination by the three cases are shown in Table 5.

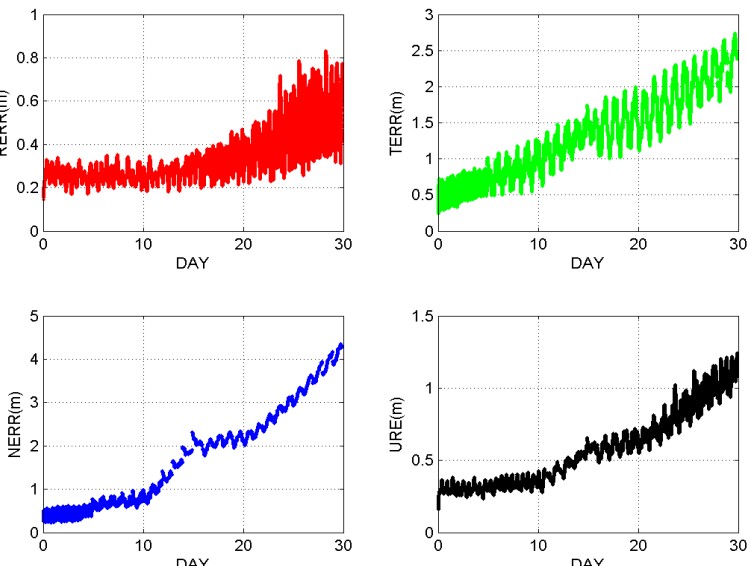

**Figure 9.** The autonomous navigation constellation average orbit error with independent constraints to correct the rotation case.

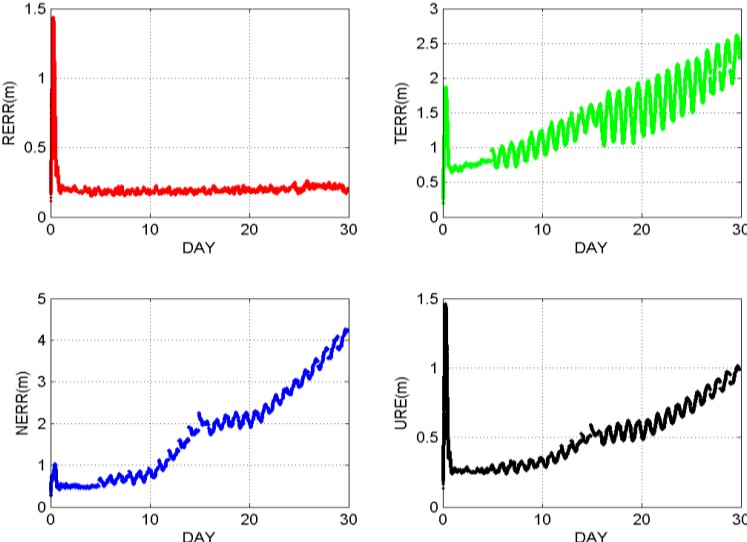

**Figure 10.** The autonomous navigation constellation average orbit error with overall constraints onboard the satellite to correct the rotation case.

**Table 5.** The autonomous navigation constellation average orbit error with different cases.

| Case | RERR (m) | TERR (m) | NERR (m) | URE (m) |
|---|---|---|---|---|
| Case 1 | 0.17 | 4.70 | 4.55 | 1.41 |
| Case 2 | 0.54 | 2.21 | 4.07 | 1.09 |
| Case 3 | 0.17 | 2.14 | 4.02 | 0.94 |

For the 24 BDS-3 MEO satellites, the orbit-only URE is expressed as:

$$URE = \sqrt{0.96 * \text{RERR}^2 + 0.04 * \left(\text{TERR}^2 + \text{NERR}^2\right)} \tag{23}$$

As can be seen, without rotation correction, the average RMS of the RERRs for the entire constellation in 30 days was approximately 0.17 m, the TERR was 4.70 m, the NERR was 4.55m, and the orbit accuracy was approximately 1.41 m. After adopting the algorithm for independent satellite constraints to carry out rotation correction, the average RMS of RERR was 0.54 m, the RMS of TERR was 2.21 m, the RMS of NERR was 4.07 m, and the orbit accuracy of was 1.09 m. By adopting the algorithm for global satellite constraints, the average RMS of RERR was 0.17 m, and the NERR and TERR were 2.14 m and 4.02 m, respectively, and the accuracy of orbit determination for the entire constellation was 0.72 m. Among them, the 30-day accuracy in autonomous orbit determination of without rotation correction was the worst, and the accuracy under the algorithm for global constraints on satellites was the best. The orbit accuracy of the three cases of autonomous orbit determination deteriorated with time gradually, because the errors created by UT1-UTC prediction in autonomous orbit determination cannot be eliminated relying on only the observed values of inter-satellite links but without external reference constraints. The orbit error trend was basically consistent with the prediction errors of UT1-UTC in Figure 4.

Compared with Case 1, the algorithm for independent satellite constraints significantly decreased TERR and NERR in the orbit determination. However, the RERR showed an increasing trend. The reason is that the rotation parameters of each satellite in the independent constraint algorithm are calculated separately, which caused variations in the geometry configuration of the inter-satellite ranging network [11]. When the $\Omega$ variations of two satellites were the same, the inter-satellite distance was not to be changed. Conversely, a change in the inter-satellite distance was unavoidable. Table 4 shows the $\Omega$ error of the predicted orbit for each satellite. It can be seen that the $\Omega$ of each satellite was different and it would cause errors in the inter-satellite distance, which would indirectly lead to orbital RERR. Compared with the algorithm for independent satellite constraints, the algorithm for global satellite constraints achieved a smaller improvement in the tangential and normal directions, but it constrained the RERR well. This result illustrates that the algorithm for independent satellite constraints did not destroy the configuration of autonomous orbit determination. Therefore, it would not lead to an increasing orbit RERR.

The following conclusions were summarized through this study: (1) Under the premise of no external constraints, the accuracy of autonomous orbit determination deteriorates with time. (2) Both algorithms for rotation correction can weaken the influence of the overall constellation rotation. (3) The algorithm for independent satellite constraints will affect the integrity of the constellation configuration as a rigid body to a certain extent, which may lead to an increasing RERR. However, the algorithm for global satellite constraints will not destroy the constellation configuration. (4) Among the three cases, the 30-day accuracy in autonomous orbit determination using the algorithm for global satellite constraints was the best.

## 5. Conclusions

In this paper, the models for constellation rotation errors during autonomous orbit determination were analyzed theoretically. Aiming at the drawbacks of the algorithm for independent satellite constraints in distributed autonomous orbit determination, an algorithm for global satellite constraints was proposed. Based on the inter-satellite observation data for the 24 MEOs in the BDS-3, BDS-3 autonomous orbit determination experiments were designed under three schemes for correcting the rotation and the orbit determination results were assessed. The results are described as follows:

(1) The BDS-3 inter-satellite ranging data show satisfactory continuity, and the average number of established links for a single satellite was approximately 14.4. The geometric configuration of inter-satellite link establishment was good, with an average PDOP of approximately 0.99.

(2) Among three rotation-correction cases, the algorithm for independent satellite constraints and the global satellite constraints constrained the constellation rotation on the X and the Y axes better. However, this method did not improve the Z-axis rotation errors caused by the prediction errors in UT1-UTC.

(3) Under the three rotation-correction schemes, distributed autonomous orbit determination processing can be carried out. Compared with precise ephemeris, it shows that without external constraints, the accuracy of autonomous orbit determination deteriorates with time. With no correction algorithm applied, the average RMS of the 30-day orbit URE of autonomous orbit determination was 1.41 m. Using the algorithm for independent satellite constraints, the average RMS of the 30-day orbit URE was 1.09 m. Using the algorithm for global satellite constraints, the average RMS of the 30-day orbital URE was 0.94 m, and this scheme achieves stable and reliable autonomous orbit determination results.

(4) Both the algorithm for independent satellite constraints and the algorithm for global satellite constraints can weaken the influence of the overall constellation rotation. The former will affect the integrity of the constellation configuration as a rigid body to a certain extent, while the later solves this problem perfectly. Among the three schemes, the 30-day accuracy of autonomous orbit determination using the algorithm for global constraints on satellites was the best.

The work conducted and the conclusions obtained in this paper provide some reference for improving the accuracy of autonomous orbit determination. However, in a case where the ground reference benchmarks are missing, the constellation rotation errors of autonomous orbit determination caused by UT1-UTC predictions using the inter-satellite link ranging data cannot be eliminated. To follow up, further research and performance analysis on autonomous orbit determination technology for the navigation constellation based on anchoring support should be carried out.

**Author Contributions:** W.Z., Z.L. and W.L. conceived and designed the experiments; W.Z., H.C. and Z.L. performed the experiments and analyzed the data; W.Z., H.C. and Z.L. wrote the paper; C.T., X.H. and W.L. reviewed the paper. All authors have read and agreed to the published version of the manuscript.

**Funding:** This research received no external funding.

**Data Availability Statement:** The data presented in this study are available on request from Beijing Institute of Tracking and Telecommunications Technology (BITTT). The data are not publicly available due to the confidentiality of the data.

**Conflicts of Interest:** The authors declare no conflict of interest.

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
