# Peer review of "Research on the Rotational Correction of Distributed Autonomous Orbit Determination in the Satellite Navigation Constellation"

_remotesensing, doi:10.3390/rs14143309_

Round 1
Reviewer 1 Report
Please see the attachment.

Author Response
Dear Reviewer
We would like to thank you for your careful reading, helpful comments, and constructive suggestions, which has significantly improved the presentation of our manuscript.We have carefully considered all comments from the reviewers and revised our manuscript accordingly. The manuscript has also been double-checked, and the typos and grammar errors we found have been corrected. In the following section, we summarize our responses to each comment from the reviewers. We believe that our responses have well addressed all concerns from the reviewers. We hope our revised manuscript can be accepted for publication.

Reviewer 2 Report
This paper addresses the rotational corrections of BDS-3 autonomous orbit determination using inter-satellite ranging observations. The methods and results are quite new and interesting to the GNSS community. Therefore, I would recommend accepting for publication after conducting minor revisions. My specific comments are given below:
1. As far as I know, GPS also conducts some experiments on intersatellite link, and some results have been published, which should also be included in the Introduction. Furthermore, what are the current methods & work for controlling rotation errors?
2. P2 Line 74-76: the citation format can be improved.
3. P4 Line 154: what is H?
4. P9 Figure 4, top-right “EOP data” in red?? The font size is too small to see it clear. This figure needs to be replaced.
5. P14 Line 398: it shows..
6. I would recommend also including full descriptions of case 1-3 in the Figure and table captions for better understanding. It will help readers to get general ideas of the paper before reading the whole paper thoroughly.
7. In figure 9-10, the radial error for case 1 and case 3 is quite small and stable over time…What is the equation for calculating URE? As far as I know, the radial component account for the major part of the URE for MEO satellites.
Author Response
Dear reviewer
We would like to thank you for your careful reading, helpful comments, and constructive suggestions, which has significantly improved the presentation of our manuscript.We have carefully considered all comments from the reviewers and revised our manuscript accordingly. The manuscript has also been double-checked, and the typos and grammar errors we found have been corrected. In the following section, we summarize our responses to each comment from the reviewers. We believe that our responses have well addressed all concerns from the reviewers. We hope our revised manuscript can be accepted for publication.

Reviewer 3 Report
# General
Increase the spacing between the equations and the text. With this simple measure you can increase the overall readibility a lot.
# Specific
Line 46: Insert proper reference here.
Page 2 (Introduction): The references format is a little bit odd. Usually, only brackets [xyz] should be written, not the full list of authors:
"In addition, Wanke Liu, Fuhong Wang, Xiaoying Gong, Jinjun Zheng, Yanling hen, Yujun Du, Weixing Zhang, and others ..."
Equation (10) / (11) / (17): What is "H"?
Page 9, lines 274 - 276: Explain case 2/3, the "adoption of an algorithm...". What type of algorithm is this? Are there any references?
Author Response

(The authors gave the same response as above.)
